# Low-Energy Transcranial Navigation-Guided Focused Ultrasound for Neuropathic Pain: An Exploratory Study

**DOI:** 10.3390/brainsci13101433

**Published:** 2023-10-08

**Authors:** Dong Hoon Shin, Seong Son, Eun Young Kim

**Affiliations:** 1Department of Neurology, Gachon University Gil Medical Center, Incheon 21565, Republic of Korea; wadada@gilhospital.com; 2Department of Neurosurgery, Gachon University Gil Medical Center, Incheon 21565, Republic of Korea; nseykim@gilhospital.com

**Keywords:** anterior cingulate cortex, deep brain stimulation, neuropathic pain, percutaneous electrical neuromodulation, ultrasonic therapy

## Abstract

Neuromodulation using high-energy focused ultrasound (FUS) has recently been developed for various neurological disorders, including tremors, epilepsy, and neuropathic pain. We investigated the safety and efficacy of low-energy FUS for patients with chronic neuropathic pain. We conducted a prospective single-arm trial with 3-month follow-up using new transcranial, navigation-guided, focused ultrasound (tcNgFUS) technology to stimulate the anterior cingulate cortex. Eleven patients underwent FUS with a frequency of 250 kHz and spatial-peak temporal-average intensity of 0.72 W/cm^2^. A clinical survey based on the visual analog scale of pain and a brief pain inventory (BPI) was performed during the study period. The average age was 60.55 ± 13.18 years-old with a male-to-female ratio of 6:5. The median current pain decreased from 10.0 to 7.0 (*p* = 0.021), median average pain decreased from 8.5 to 6.0 (*p* = 0.027), and median maximum pain decreased from 10.0 to 8.0 (*p* = 0.008) at 4 weeks after treatment. Additionally, the sum of daily life interference based on BPI was improved from 59.00 ± 11.66 to 51.91 ± 9.18 (*p* = 0.021). There were no side effects such as burns, headaches, or seizures, and no significant changes in follow-up brain magnetic resonance imaging. Low-energy tcNgFUS could be a safe and noninvasive neuromodulation technique for the treatment of chronic neuropathic pain

## 1. Introduction

Neuropathic pain is a complex chronic condition with a major impact on quality of life [1]. Neuropathic pain is defined as pain caused by a lesion or disease of the somatosensory nervous system according to the International Association of the Study of Pain. This common condition, including trigeminal neuralgia, neuropathic pain related to peripheral nerve injury, painful polyneuropathy, postherpetic neuralgia, painful radiculopathy, and central neuropathic pain, makes the patients suffer from burning, shooting, pricking, pins and needles, squeezing, freezing pain, paroxysmal pain, or cold- or touch-evoked allodynia [2,3]. 

Although the mechanism of DBS is still unclear and the outcomes are variable, DBS for neuropathic pain has gradually gained consensus approval worldwide [4,5,6]. Deep brain stimulation (DBS) has been used for more than 60 years to alleviate neuropathic pain, and the number of patients undergoing DBS is increasing [7,8]. When pain relief is not satisfactory despite sufficient medication or is associated with side effects, brain neuromodulation can be one of the strategies [9]. Several anatomical points, including the periaqueductal/periventricular gray matter and dorsal ventral nucleus/lateral nucleus of the thalamus, were recruited as targets of stimulation for contraction of the sensory component of the pain transition tract [7,10,11,12]. Stimulation of these somatosensory circuits appears to be effective in the sensory component of pain transactions [5].

The anterior cingulate cortex (ACC), which is involved in many motor and psychological functions and plays an important role in pain and empathy, has emerged as an alternative target for neuropathic pain [13,14]. Pain is a complex sensation composed of at least three components: sensory identification (pain intensity), affective (pain unpleasantness), and cognitive components [15]. Based on this concept, not all patients responded to stimulation of the primary sensory pathway, and other target sites, such as the area responsible for the emotion, were sought accordingly. In the previous study, DBS of the ACC significantly improved the pain and quality of life of patients after 6 months, and the effects persisted for up to 42 months in some patients [16,17]. 

Recently, there has been a paradigm shift towards less invasive methods of brain stimulation, and transcranial focused ultrasound (tcFUS) is one such method [18]. Because ultrasound can deliver stimulation noninvasively through the skull to the targeted deep brain, it has the potential to replace conventional invasive neuromodulation techniques [19,20]. Several studies have reported that high-energy focused ultrasound (FUS) technology can be used to reduce neuropathic pain [21,22,23].

Building upon these findings, our hypothesis posited that low-energy sonication could provide a safer means of neuromodulation without irreversible brain-tissue damage. In this exploratory clinical trial, grounded in the aforementioned research, we conducted transcranial navigation-guided focused ultrasound (tcNgFUS) using low-intensity sonication in patients with intractable neuropathic pain. Our objective was to establish the safety and efficacy of this treatment approach. 

## 2. Materials and Methods

### 2.1. Trial Design and Ethics

This was a prospective, single-center, single-arm, open-label, exploratory trial with a 3-month follow-up. 

The study was conducted in accordance with the 1964 Helsinki Declaration and its later amendments and was approved by the Institutional Review Board of our institute (GCIRB2020-030). This study was registered as a clinical trial in the Clinical Research Information Service of South Korea (KCT 0007894).

### 2.2. Aims 

This clinical trial aimed to explore the efficacy and safety of low-energy tcNgFUS for pain relief in patients with uncontrolled neuropathic pain. 

The primary endpoint was efficacy based on pain relief measured using the visual analog scale (VAS). The secondary endpoint included the effectiveness of treatment from a subjective patient’s viewpoint and improvement of daily life interference based on the Korean version of the Brief Pain Inventory (K-BPI) [24].

Safety was estimated by investigating any procedure-related adverse events or changes in follow-up brain MRI after treatment. 

### 2.3. Time Frame 

The final follow-up period was up to 3 months, and the study was conducted over 18 months.

TcNgFUS stimulation was performed according to detailed protocols for 2 weeks, three times a week. A clinical survey was conducted before the procedure and immediately, 2 weeks, and 4 weeks after the procedure. The final follow-up was conducted 3 months after the procedure for safety evaluation and follow-up brain magnetic resonance imaging (MRI) (Figure 1). 

### 2.4. Sample Size

The number of samples was calculated as follows: $$N=\frac{{\left(z_{1-\alpha /2}+z_{1-\beta }\right)}^{2}\sigma^{2}}{d^{2}}$$

Based on the results of previous clinical trials of neuromodulation [25,26,27,28], the standard deviation (σ) of the main effect variable was set to 3.5, and the difference of effect (d) was set to 3. Using the above formula, with a significance level of 5% and power of 80%, a sample size of 10 patients was calculated. Considering a dropout rate of 20%, we recruited a total of 12 patients. 

### 2.5. Indications and the Patient Population

Participants were recruited from the outpatient clinic of a neurologist or neurosurgeon specializing in spine disease, stroke, or functional brain surgery.

All potential participants were screened to determine their eligibility according to the following inclusion and exclusion criteria. The inclusion criteria were as follows: (1) adults aged 19–75 years; (2) persistent neuropathic pain for more than 6 months, as determined by International Classification of Disease-10 codes of T060, T061, T093, T913, S141, S241, S246, S341, S342, M890, M961, G500, G530, G546, G564, G631, G632, G5780, G5880, and G5881; (3) intractable pain despite a minimum of 4 weeks of medication and any intervention such as nerve block, epidural block, or spinal cord stimulation; (4) severe pain that interferes with daily life, defined as a VAS pain level ≥ 5 and pain disability index ≥ 30 [29]; (5) a skull density ratio, calculated from the Hounsfield unit value of the skull, of at least 0.4 for successful ultrasonic energy arrival [30,31]; and (6) voluntary participation and compliance with the clinical trial protocol. The exclusion criteria were as follows: (1) contraindication of MRI or computed tomography (CT) such as claustrophobia, in situ implant, pregnant, lactating, or women with childbearing potential who plan to become pregnant; (2) history of previous DBS surgery; (3) severe mental disorders such as uncontrolled depression, bipolar disorder, or alcohol abuse; (4) unstable medical conditions, such as cardiopulmonary disease, uncontrolled hypertension, immune deficiency, cancer, or coagulopathy; (5) life expectancy ≤12 months; (6) active brain lesions, such as stroke, vascular malformation, or tumor, on screening brain MRI or CT; (7) inability to maintain a stationary posture for the required amount of time during the procedure; or (8) inappropriate participation in this clinical trial according to the judgment of the researchers. 

### 2.6. Procedure Protocol

After patient selection and informed consent, all patients underwent pre-procedure brain CT using Somatom force^®^ (Siemens, Erlangen, Germany) with a 1 mm isovoxel and brain MRI using Skyra 3T^®^ (Siemens, Erlangen, Germany) with 1 mm isovoxel to confirm the skull and brain anatomical structure. All imaging was performed after the attachment of a multimodal sticker (fiducial marker) for the fusion of both CT and MRI images and the planning of image guidance. Through acquired brain image processing, the brain anatomical/functional target of the ACC area for each participant was identified, and image-guided FUS was designed [32].

TcNgFUS was performed using an NS-US100^®^ (Neurosona Corporation, Seoul, Republic of Korea). Treatment was carried out three times a week for 2 weeks (every Monday, Wednesday, and Friday). The visitation period for each treatment was 2 days within the planned date of treatment, and if the attendance rate was less than 80%, the patient would drop out. 

The stimulation target was Brodmann area 24 (dorsal ACC), which was a white matter located 20 mm posterior to the anterior tip of the frontal horns of the lateral ventricles [33]. During stimulation, the deepest margin was the corpus callosum, and the shallowest margin was the cingulum bundle [16]. 

Based on the acquired brain anatomical and functional images, FUS was applied noninvasively to the bilateral ACC of the participant. The transducer with a distance to target of 70 cm, cross-sectional area of target of 38.48 mm^2^, and volume of target of 5280 mm^3^ was used. To increase the transmittance of ultrasound across the skin, ultrasound was applied after the hydrogel had spread to the scalp of the participant (Figure 2).

Low-energy FUS had a frequency of 250 kHz, spatial-peak temporal-average intensity of less than 0.72 W/cm^2^, and tone burst duration of 5–10 ms. Stimulation was performed with a 50–70% duty cycle (the rate occupied by an ultrasonic donation per second). The total stimulation time, including the rest period, did not exceed 30 min in all patients. Additionally, the device was used within a range that does not exceed a Mechanical Index of 1.9 (in situ peak rarefaction pressure less than 0.95 MPa according to the Food and Drug Administration safety standards for ultrasonic imaging devices). 

Drug treatment was maintained as usual as the pre-procedure state for all patients.

### 2.7. Outcome Assessment 

Demographic data, including age and sex, and baseline characteristics, including diagnosis and symptom duration, were recorded. 

Clinical data, including VAS and K-BPI, were obtained before the procedure, immediately after completion of the procedure, and at 2 and 4 weeks after the procedure in all patients. The degree of current pain, average pain during the last 24 h, and maximum pain during the last 24 h were surveyed using VAS scores ranging from 0 to 10 points. Pain-related quality of life was assessed using K-BPI [24]. The sum of daily life interference and effectiveness of treatment from the patient’s viewpoint was collected. 

Procedure-related complications, such as the burning of the scalp, headache, seizure, and aggravation of pain, were observed during the follow-up period of up to 3 months after the procedure in all patients. In addition, a post-procedure brain MRI was performed 3 months after the procedure to check for any adverse events in the structure around the stimulated target. 

### 2.8. Statistical Analysis

Data management and statistical analyses were performed using SPSS (version 27.0; IBM Corporation, Armonk, NY, USA). The non-parametric Friedman test, one-way analysis of variance (ANOVA), and paired *t*-test were used according to the characteristics of the factors. 

The results were expressed as mean ± standard deviation, median with range, or mean and corresponding 95% confidence interval (CI) depending on whether the data were normally distributed or not. Statistical significance was set at *p*-values < 0.05. 

## 3. Results

### 3.1. Demographic Data and Baseline Characteristics 

Among the 12 registered patients, one patient was excluded because of withdrawal of consent due to simple change of mind during follow-up. The 11 final study participants comprised 6 men and 5 women, with an overall mean age of 60.55 ± 1.18 years (range, 44–81). Patients were classified based on their diagnosis as four patients with spinal cord injury, three with compressive myelopathy, three with post spinal surgery syndrome, and one with cauda equina syndrome. The median symptom duration was 4.33 (range, 1.42–13.17) years (Table 1).

### 3.2. Clinical Outcome Related to Pain Degree

The median current pain VAS at pre-procedure was 10.0 (range, 6.0–10.0), and this decreased to 8.0 (range, 3.0–10.0) immediately post-procedure, 7.0 (range, 3.0–900) at 2 weeks post-procedure, and 7.0 (range, 4.0–9.0) at 4 weeks post-procedure (*p* = 0.039, non-parametric Friedman test). The pre-procedure to 2 weeks post-procedure and pre-procedure to 4 weeks post-procedure differences in current VAS scores were significant (mean difference 1.91 [95% CI, 0.36–3.45] and 2.09 [95% CI, 0.64–3.55], *p* = 0.048 and 0.021, respectively, non-parametric Friedman test); however, the difference during follow-up after the procedure was not significant (Table 2, Figure 3).
Figure 3Change of current pain degree according to the visual analog scale. The median average pain VAS at pre-procedure was 8.5 (range, 5.0–10.0), and this decreased to 6.0 (range, 2.0–9.0) immediately post-procedure, 6.0 (range, 2.0–9.0) at 2 weeks post-procedure, and 6.0 (range, 4.0–9.0) at 4 weeks post-procedure (*p* = 0.070, non-parametric Friedman test). The pre-procedure to 4 weeks post-procedure difference in average pain VAS was significant (mean difference 1.50 [95% CI, 0.41–2.59], *p* = 0.027, non-parametric Friedman test); however, other differences between each period were not significant (Table 2, Figure 4). * *p* < 0.05 and ⸰ are lables that are out of range.
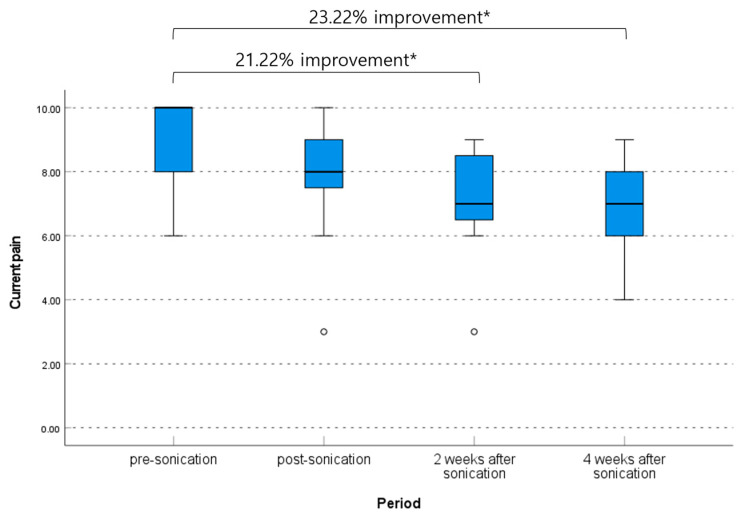


The median maximum VAS at pre-procedure was 10. 0 (range, 6.0–10.0), and this decreased to 9.0 (range, 3.0–9.0) immediately post-procedure, 9.0 (range, 3.0–9.0) at 2 weeks post-procedure, and 8.0 (range, 4.0–9.0) at 4 weeks post-procedure (*p* = 0.012, non-parametric Friedman test). The pre-procedure to 2 weeks post-procedure and pre-procedure to 4 weeks post-procedure difference in maximum VAS scores were significant (mean difference 1.50 [95% CI, 0.83–2.92] and 1.77 [95% CI, 0.56–2.99], *p* = 0.032 and 0.008, respectively, non-parametric Friedman test); however other differences between each period were not significant (Table 2, Figure 5). 

### 3.3. Clinical Outcome Related to Daily Life Quality 

The mean of the sum of interference of daily life based on BPI was improved significantly from 59.00 ± 11.66 at pre-procedure to 50.09 ± 16.13 immediately post-procedure, 55.73 ± 9.75 at 2 weeks post-procedure, and 51.91 ± 9.8 at 4 weeks post-procedure (*p* = 0.318, one-way ANOVA). The pre-procedure to immediately post-procedure and pre-procedure to 4 weeks post-procedure differences in sum of interference of daily life were significant (the mean difference 8.91 [95% CI, 0.87–16.95] and 7.09 [95% CI, 1.29–12.89], *p* = 0.033 and 0.021, respectively, paired *t*-test) (Table 3, Figure 6). 

On the other hand, the subjective pain-relieving effect of treatment according to K-BPI did not significantly improve, from 60.0 (range, 0.0–90.0%) at pre-procedure, and 50.0 (range, 0.0–90.0%) after the procedure (*p* = 0.429, non-parametric Friedman test) (Table 3). 

### 3.4. The Complication and Follow-Up Brain MRI Findings 

No procedure-related complications, such as a burning sensation of the scalp, headache, or seizure, were observed during the follow-up period. Additionally, no specific changes in the structure of the target, such as inertial cavitation, edema, or hemorrhage, were observed after sonication according to the follow-up brain MRI. 

## 4. Discussion

The ACC has been revealed as a key structure of the affective component among three integrated aspects including sensory, affective, and cognitive aspects of chronic neuropathic pain. Anatomically, the ACC is a part of the cingulate gyrus and consists of Brodmann areas 24, 25, 32, and 33 [33]. According to previous theory, the ACC receives afferent inputs from the midline nuclei of the thalamus, such as anteromedial, paraventricular, paracentral, central and centrolateral, reuniens, parafascicular, limitans, mediodorsal, and ventral anterior nuclei. Ref. [13] Several tracts then connect the ACC to the motor system (premotor and motor cortices, spinal cord), cognition-related areas (cingulate motor areas in the cingulate sulcus and nociceptive cortex), and the limbic system related to emotion (amygdala, insula, hypothalamus, ventral striatum, and ventromedial prefrontal cortex) [34,35,36].

The stimulation of the ACC was supposed to improve the affective component of pain, whereas stimulation of the thalamus or periventricular gray region was shown to reduce the intensity of pain [37,38]. Consequently, the description of the patient’s condition after ACC stimulation was “less frustrating”, “less distressing”, and “less worried” [16]. Since the first clinical outcome after stimulation of the dorsal ACC to relieve whole-body pain by cervical spinal cord injury was reported in 2007, much clinical research has suggested the potential of ACC targeting to improve quality of life in patients with chronic pain [39,40,41,42].

However, DBS requires the invasive surgical implantation of electrodes and causes fear/inconvenience and surgery-related complications. According to a previous study, complication rates were reported with a remarkable probability, including 16.7% for infection, 8.3% for broken lead, and 16.7% for seizure, after DBS surgery of the ACC for neuropathic pain [16]. Moreover, recent pharmacotherapy provides a sufficient level of pain relief in approximately 30–40% of patients. Accordingly, the practical efficacy of conventional DBS should be considered and carefully determined based on proper patient selection. 

To overcome the limitations of invasive implantation surgery, several new trials based on advanced technology have been developed in the last decade, and tcFUS is one of the most promising methods. FUS is a precise, noninvasive, and radiation-free technique in which ultrasound single waves reach the target without acoustic reflection, refraction, and distortion through trajectory [43,44,45,46]. Compared to other noninvasive brain stimulation techniques, such as magnetic or electric stimulations, FUS can stimulate any target neural structure for pain relief with a delicate spatial resolution [47]. In addition, FUS offers a variety of biological intensities, which lead to different treatment effects, from reversible neuromodulation by low-intensity and low-frequency ultrasound to irreversible tissue damage by high-intensity ultrasound [48,49]. 

Classic high-energy FUS has two main bioeffects: the local thermal effect and inertial cavitation. Protein denaturation and cell death, which result from local heat generation, and inertial cavitation, which is caused by the collapse of gas bubbles owing to the pressure exerted by the ultrasonic field, induce irreversible tissue ablation [48]. However, irreversible tissue ablation using high-energy FUS has some procedure-related complications or side effects on neighboring structures [22,44,50].

Accordingly, there has been a growing interest in the reversible biological effects of FUS, and many researchers have tried to improve the neurological and histological effects of low-energy FUS [44,51]. The hypothesis of the biomechanism of low-energy FUS is the modification of neuronal membranes through the activation of mechanosensitive voltage-gated sodium and calcium channels or neurotransmitter receptors [52]. In addition, another bioeffect of low-energy FUS is to induce pore formation in the cellular membrane, which changes membrane permeability [53]. Recently, some studies have found that it is possible to modulate neural circuits without irreversible tissue damage using non-ablative doses [54,55,56].

TcFUS has been approved for the treatment of refractory essential tremors and is being studied for other neurological indications, including dyskinesias and tremors in Parkinson’s disease, dystonia, obsessive-compulsive disorder, epilepsy, brain tumors, and traumatic brain injury [57,58,59,60]. Recently, the safety and efficacy of FUS in neuropathic pain have been discovered [8,49,50,61]. Moreover, although currently in the research phase, low-energy FUS has potentially reversible and tissue-selective effects [21,23,62,63,64]. 

MRI-based tcNgFUS device is an advanced technology using low-energy FUS. According to the standards of the Ministry of Food and Drug Safety, ultrasonic waves should be used within a range of less than 3 W/cm^2^ to prevent heat generation or inertial cavitation [47,65]. FUS with a low frequency of <500 kHz and low spatial-peak temporal-average of <3 W/cm^2^ generates a pressure wave of mechanical energy, unlike high-intensity FUS, which generates heat [45]. 

The patients who underwent low-energy FUS in this clinical trial showed a mean current pain relief profile across time, from 21.22% (from 9.00 to 7.09) at 2 weeks to 2.54% (from 7.09 to 6.91) at 4 weeks after treatment. The improvement in the maximum pain during the last 24 h was similar, as 15.94% (from 9.41 to 7.91) at 2 weeks and 3.41% (from 7.91 to 7.64) at 4 weeks. Furthermore, an improvement in interference with daily life based on BPI was observed to be as significant as 24.45% (from 59.00 to 51.91) at 4 weeks after the procedure. Because the primary effect of low-energy FUS is not pain alleviation like high-energy FUS, patient satisfaction based on subjective pain-relieving effect may appear insignificant. However, patients tend to describe that pain is still present but that it is less stressful. In other words, the improvement in pain and life quality is statistically significant, although the degree of improvement is slightly lower than the results of the previous literature at 30–40% improvement [16,50,57,61]. 

The present study had several limitations. First, owing to its single-arm design, direct comparison with the sham group was not possible. Therefore, the placebo effect cannot be disregarded. Second, the number of enrolled patients was small. Nonetheless, regarding the purpose of this study as a preliminary exploration, it provides sufficient evidence to prove the safety and efficacy of low-energy tcNgFUS. Third, the follow-up period was relatively short. Improvement in early phase can diminish over time owing to reversible lesioning and the possibility of brain plasticity [66]. A fourth limitation is the absence of objective indicators for evaluating treatment effect. Previous research in both animal and human experiments has explored objective measures, including action potentials, evoked electroencephalographic responses, and tactile sensations experienced by subjects during FUS stimulation, to validate the efficacy of FUS. However, specific objective parameters or biomarkers capable of confirming the results after FUS treatment in patients with neuropathic pain remain unidentified [55,56,67]. Consequently, the assessment of FUS effects relied solely on the patient-reported outcomes. To overcome this limitation, it is crucial to conduct additional research focused on identifying objective indicators in conjunction with ongoing clinical surveys.

The trend of the results of this study was not superior to that in the previous literature based on DBS or high-energy FUS for neuropathic pain. However, we believe that the obtained results are valuable because this is the first pioneering study of low-energy FUS for chronic neuropathic pain. We suggest that further large-scale, randomized controlled trials with long-term follow-up for direct comparison between case and control groups should be conducted to confirm our results.

## 5. Conclusions

This prospective exploratory study showed that low-energy tcNgFUS could be used effectively in patients with chronic neuropathic pain. Low-energy FUS avoids procedure-related complications and irreversible brain-tissue damage. In addition, portable real-time navigation guidance is accurate for planning targeting and comfortable for the patient. However, further clinical investigations are necessary to confirm its clinical safety and long-term efficacy.

## Figures and Tables

**Figure 1 brainsci-13-01433-f001:**
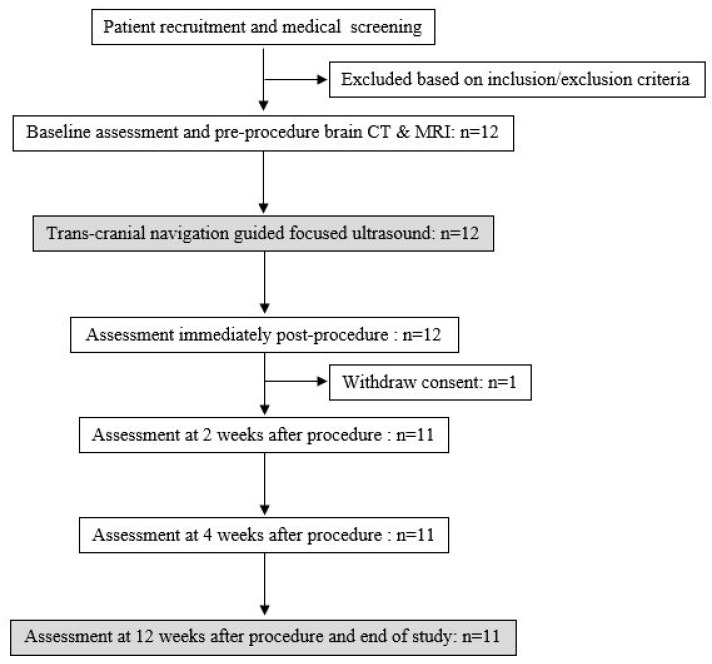
Diagram of participant recruitment and study design.

**Figure 2 brainsci-13-01433-f002:**
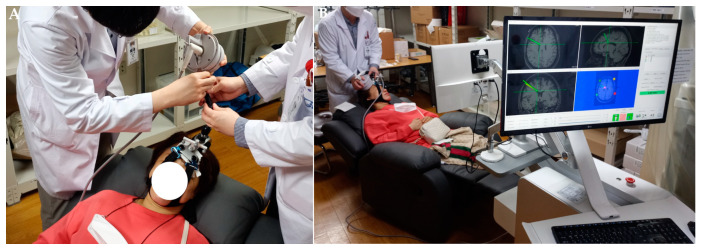
The process of administering low-energy focused ultrasound to a patient. (**A**) The fiducial marker and head gear are worn on the patient’s face and head, and the transducer is set. (**B**) The anterior cingulate cortex is targeted under the navigation guidance based on magnetic resonance imaging performed before the procedure. Once correct targeting is confirmed, planned sonication is performed.

**Figure 4 brainsci-13-01433-f004:**
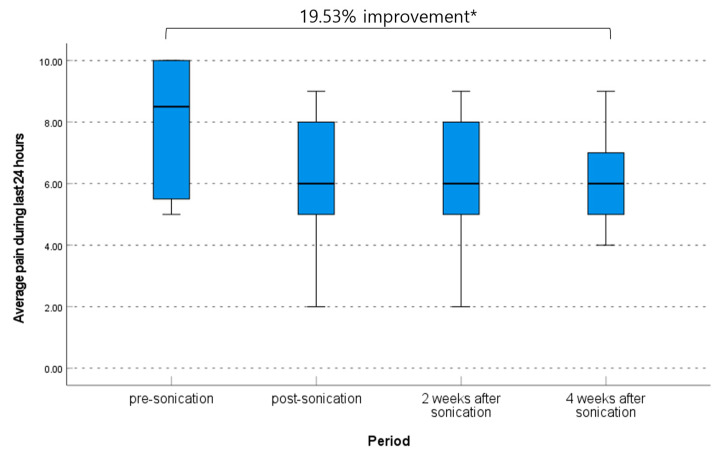
Change in average pain degree within the last 24 h according to the visual analog scale. * *p* < 0.05.

**Figure 5 brainsci-13-01433-f005:**
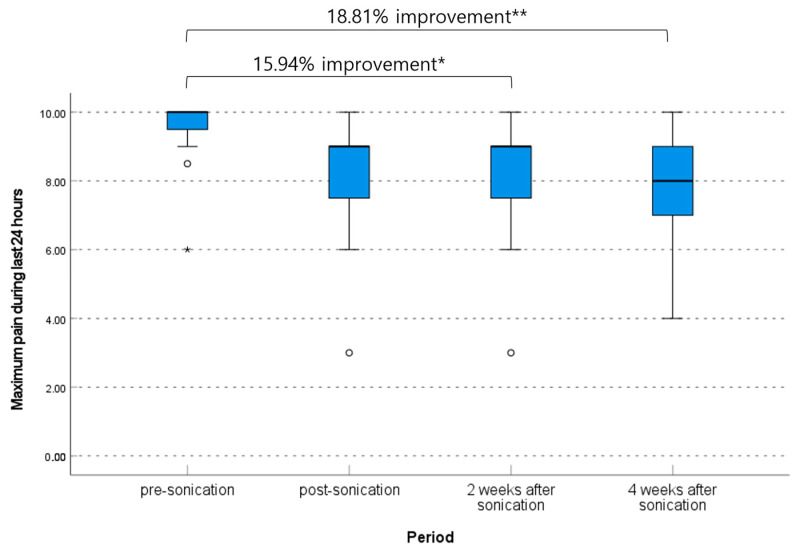
Change of maximum pain degree within the last 24 h according to the visual analog scale. * *p* < 0.05, ** *p* < 0.01, and ⸰ are lables that are out of range.

**Figure 6 brainsci-13-01433-f006:**
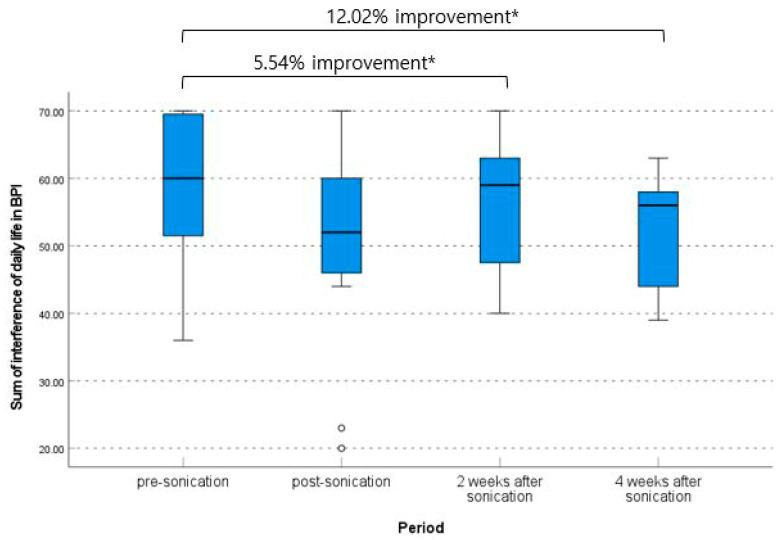
Change of sum of interference of daily life based on brief pain inventory. * *p* < 0.05 and ⸰ are lables that are out of range.

**Table 1 brainsci-13-01433-t001:** Demographic data and baseline characteristics.

Characteristics	Patients (*n* = 11)
Age (years, mean ± SD)	60.55 ± 13.18 (range, 44.00–81.00)
Men/Women	6/5
The median duration of symptoms (years)	4.33 (range, 1.42–13.17)
Location of pain: Right/Left/Both	4/1/6
Diagnosis	
Spinal cord injury	4
Compressive myelopathy	3
Post spinal surgery syndrome	3
Cauda equina syndrome	1

SD, standard deviation.

**Table 2 brainsci-13-01433-t002:** Clinical outcomes related to pain.

Characteristics	Values	*p* Value
Current pain		0.039 ^†^
Initial	10.0 (range, 6.0–10.0)	
Immediately post-procedure	8.0 (range, 3.0–10.0)	
2 weeks	7.0 (range, 3.0–9.0)	
4 weeks	7.0 (range, 4.0–9.0)	
Δ Current pain		
Initial—Immediately post-procedure	1.18 (95% CI, −0.40–2.77)	0.509 ^†^
Initial—2 weeks	1.91 (95% CI, 0.36–3.45)	0.048 ^†^
Initial—4 weeks	2.09 (95% CI, 0.64–3.55)	0.021 ^†^
Immediately post-procedure—2 weeks	0.73 (95% CI, −0.13–1.58)	0.186 ^†^
Immediately post-procedure—4 weeks	0.91 (95% CI, 0.09–1.73)	0.099 ^†^
2 weeks–4 weeks	0.18 (95% CI, −0.54–0.91)	0.741 ^†^
Average pain during the last 24 h		0.070 ^†^
Initial	8.5 (range, 5.0–10.0)	
Immediately post-procedure	6.0 (range, 2.0–9.0)	
2 weeks	6.0 (range, 2.0–9.0)	
4 weeks	6.0 (range, 4.0–9.0)	
Δ Average pain		
Initial—Immediately post-procedure	1.50 (95% CI, −0.21–3.21)	0.079 ^†^
Initial—2 weeks	1.59 (95% CI, −0.05–3.23)	0.056 ^†^
Initial—4 weeks	1.50 (95% CI, 0.41–2.59)	0.027 ^†^
Immediately post-procedure—2 weeks	0.91 (95% CI, −0.27–0.45)	0.588 ^†^
Immediately post-procedure—4 weeks	(95% CI, −1.34–1.34)	1.000 ^†^
2 weeks–4 weeks	−0.91 (95% CI, −1.42–1.23)	0.882 ^†^
Maximum pain during the last 24 h		0.012 ^†^
Pre-procedure	10.0 (range, 6.0–10.0)	
Immediately post-procedure	9.0 (range, 3.0–10.0)	
2 weeks	9.0 (range, 3.0–10.0)	
4 weeks	8.0 (range, 4.0–10.0)	
Δ Maximum pain		
Initial—Immediately post-procedure	1.41 (95% CI, −0.38–2.86)	0.069 ^†^
Initial—2 weeks	1.50 (95% CI, 0.83–2.92)	0.032 ^†^
Initial—4 weeks	1.77 (95% CI, 0.56–2.99)	0.008 ^†^
Immediately post-procedure—2 weeks	0.91 (95% CI, −0.54–0.71)	0.741 ^†^
Immediately post-procedure—4 weeks	0.36 (95% CI, −0.65–1.37)	0.409 ^†^
2 weeks–4 weeks	0.27 (95% CI, −0.58–1.13)	0.620 ^†^

CI, confidence interval. ^†^ non-parametric Friedman test.

**Table 3 brainsci-13-01433-t003:** Clinical outcomes related to daily life quality.

Characteristics	Values	*p* Value
Sum of interference of daily life in BPI (mean ± SD)		0.318 ^†^
Pre-procedure	59.00 ± 11.66	
Immediately post-procedure	50.09 ± 16.13	
2 weeks	55.73 ± 9.75	
4 weeks	51.91 ± 9.18	
Δ Sum of interference of daily life in BPI		
Initial—Immediately post-procedure	8.91 (95% CI, 0.87–16.95)	0.033 ^‡^
Initial—2 weeks	3.27 (95% CI, −1.97–8.51)	0.194 ^‡^
Initial—4 weeks	7.09 (95% CI, 1.29–12.89)	0.021 ^‡^
Immediately post-procedure—2 weeks	−5.64 (95% CI, −12.25–0.98)	0.087 ^‡^
Immediately post-procedure—4 weeks	−1.82 (95% CI, −8.71–5.08)	0.570 ^‡^
2 weeks–4 weeks	3.82 (95% CI, −0.17–7.80)	0.058 ^‡^
The subjective pain-relieving effect of procedure (%)		0.429 ^§^
Pre-procedure	60.0 (range, 0.0–90.0)	
Immediately post-procedure	50.0 (range, 0.0–90.0)	
2 weeks	50.0 (range, 0.0–80.0)	
4 weeks	50.0 (range, 0.0–80.0)	
Δ Pain-relieving effect (%)		
Initial—Immediately post-procedure	8.2 (95% CI, −4.1–20.5)	0.172 ^§^
Initial—2 weeks	0.9 (95% CI, −11.6–20.5)	0.783 ^§^
Initial—4 weeks	3.6 (95% CI, −5.5–12.8)	0.395 ^§^
Immediately post-procedure—2 weeks	−7.2 (95% CI, −17.2–2.7)	0.121 ^§^
Immediately post-procedure—4 weeks	−4.5 (95% CI, −14.7–5.6)	0.357 ^§^
2 weeks–4 weeks	2.7 (95% CI, −5.2–10.7)	0.453 ^§^

CI, confidence interval; BPI, brief pain inventory; SD, standard deviation. ^†^ one-way analysis of variance, ^‡^ paired *t*-test, ^§^ non-parametric Friedman test.

## Data Availability

The data presented in this study are available on request from the corresponding author. The data are not publicly available due to issues of patient privacy.

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
