# Peer review of "Low-Energy Transcranial Navigation-Guided Focused Ultrasound for Neuropathic Pain: An Exploratory Study"

_brainsci, 2023, doi:10.3390/brainsci13101433_

Round 1

Reviewer 1 Report

The following is the summary of the present manuscript: Neuromodulation using high-energy focused ultrasound (FUS) has recently been developed for various neurological disorders, including tremors, epilepsy, and neuropathic pain. We investigated the safety and efficacy of low-energy FUS for patients with chronic neuropathic pain. We conducted a prospective single-arm trial with 3-month follow-up using new transcranial navigation-guided focused ultrasound (tcNgFUS) technology to stimulate the anterior cingulate cortex. Eleven patients underwent FUS with a frequency of 250 kHz and spatial-peak temporal-average intensity of 0.72 W/cm2. A clinical survey based on the visual analog scale of pain and brief pain inventory (BPI) was performed during study period. The average age was 60.55±13.18 years-old, with a male-to-female ratio of 6:5. The median current pain decreased from 10.0 to 7.0 (p=0.021), median average pain decreased from 8.5 to 6.0 (p=0.027), and median maximum pain decreased from 10.0 to 8.0 (p=0.008) at 4 weeks after treatment. Additionally, the sum of daily life interference based on BPI was improved from 59.00±11.66 to 51.91±9.18 (p=0.021). There were no side effects such as burns, headaches, or seizures, and no significant changes in follow-up brain magnetic resonance imaging. Low-energy tcNgFUS could be a safe and noninvasive neuromodulation.

The study is well written. The main drawback is the limited number of the participants. Concerning the novelty of the present study, I believe it deserves publication. There are some minor suggestions:

First, the hypothesis of the study should be better described in the introduction.

Second, in Figure 2, there are two parts. Please label the subgraph with alphabetic letters to ease the reading.

Third, in Table portion, please indicate how the continuous variables are presented, like mean+-SD.  

Reviewer 2 Report

Neuropathic pain caused by a lesion or disease of the somatosensory nervous system is a common chronic pain condition with major impact on quality of life. Examples include trigeminal neuralgia, painful polyneuropathy, postherpetic neuralgia, and central poststroke pain. Most patients complain of an ongoing or intermittent spontaneous pain of, for example, burning, pricking, squeezing quality, which may be accompanied by evoked pain, particular to light touch and cold. Ectopic activity in, for example, nerve-end neuroma, compressed nerves or nerve roots, dorsal root ganglia, and the thalamus may in different conditions underlie the spontaneous pain. Evoked pain may spread to neighboring areas, and the underlying pathophysiology involves peripheral and central sensitization.

This prospective exploratory study showed that low-energy tcNgFUS could be used  effectively in patients with chronic neuropathic pain. Low-energy FUS avoids procedure related complications and irreversible brain-tissue damage. In addition, portable real-time navigation guidance is accurate for planning targeting and comfortable for patient. However, further clinical investigations are necessary to confirm its clinical safety and long term efficacy. That's why the relevance of the work and its novelty are beyond doubt. The work produces the positive impression. The authors outline the research protocols in detail. But there are several questions for the authors.

The present study have several limitations:  

- owing to its single-arm design, direct  comparison with the sham group was not possible. Therefore, the placebo effect cannot  be disregarded;

- the number of enrolled patients was small;

- the follow-up period was  relatively short. Improvement in early phase can diminish over time owing to reversible  lesioning and the possibility of brain plasticity;

- the trend of the results of this study was not superior to that in the previous literature  based on DBS or high-energy FUS for neuropathic pain.

But, these remarks are not a fatal flaw in the paper  and the authors suggest that further large-scale, randomized controlled  trials with long-term follow-up for direct comparison between case and control groups  should be conducted to confirm their results.

Articles that can be supplemented with References:

Bachu VS, Kedda J, Suk I, Green JJ, Tyler B. High-Intensity Focused Ultrasound: A Review of Mechanisms and Clinical Applications. Ann Biomed Eng. 2021;49(9):1975-1991. doi:10.1007/s10439-021-02833-9

Giugno A, Maugeri R, Graziano F, et al. Restoring Neurological Physiology: The Innovative Role of High-Energy MR-Guided Focused Ultrasound (HIMRgFUS). Preliminary Data from a New Method of Lesioning Surgery. Acta Neurochir Suppl. 2017;124:55-59. doi:10.1007/978-3-319-39546-3_9

Diaz MJ, Root KT, Beneke A, Penev Y, Lucke-Wold B. Neurostimulation for Traumatic Brain Injury: Emerging Innovation. OBM Neurobiol. 2023;7(1):161. doi:10.21926/obm.neurobiol.2301161

Kim ES, Chang SY. Patch Clamp Technology for Focused Ultrasonic (FUS) Neuromodulation. Methods Mol Biol. 2022;2393:657-670. doi:10.1007/978-1-0716-1803-5_35

Possibly, the «Discussion» or «Introduction» should refer to studies in animal models. For example,

Vion-Bailly J, Suarez-Castellanos IM, Chapelon JY, Carpentier A, N'Djin WA. Neurostimulation success rate of repetitive-pulse focused ultrasound in an in vivo giant axon model: An acoustic parametric study. Med Phys. 2022;49(1):682-701. doi:10.1002/mp.15358

Yoo SH, Croce P, Margolin RW, Lee SD, Lee W. Pulsed focused ultrasound changes nerve conduction of earthworm giant axonal fibers. Neuroreport. 2017;28(4):229-233. doi:10.1097/WNR.0000000000000745

Maladaptive structural changes and a number of cell-cell interactions and molecular signaling underlie the sensitization of nociceptive pathways. These include alteration in ion channels, activation of immune cells, glial-derived mediators, and epigenetic regulation. The major classes of therapeutics include drugs acting on α2δ subunits of calcium channels, sodium channels, and descending modulatory inhibitory pathways. Would it be interesting to study changes at the molecular level in patients, as well as find information in animal models? How do blood parameters and markers of the state of neurons change, for example? Can you speculate on this topic?

These remarks are not a fatal flaw in the paper and the work can be published after minor revision.
